DEVOUR: Deleterious Variants on Uncovered Regions in Whole-Exome Sequencing

Türk Erdem erdemturk@mu.edu.tr 1 2
Ayaz Akif 3
Yüksek Ayhan 1
Süzek Barış E. 1 2
1 Department of Computer Engineering, Muğla Sıtkı Koçman University , Muğla , Turkey
2 Bioinformatics Graduate Program, Muğla Sıtkı Koçman University , Muğla , Turkey
3 Department of Medical Genetics, School of Medicine, İstanbul Medipol University , İstanbul , Turkey
Folkersen Lasse
Electronic publication date: 2023 Sep 15
Publication date: 2023
Volume: 11
Electronic Location ID: e16026
Received 2023 May 9; Accepted 2023 Aug 13
Copyright: ©2023 Türk et al.
Copyright year: 2023
Copyright holder: Türk et al.
License: This is an open access article distributed under the terms of the Creative Commons Attribution License, which permits unrestricted use, distribution, reproduction and adaptation in any medium and for any purpose provided that it is properly attributed. For attribution, the original author(s), title, publication source (PeerJ) and either DOI or URL of the article must be cited.
License URL: https://creativecommons.org/licenses/by/4.0/

Keywords: Next-generation sequence (NGS) analysis, Whole-exome sequencing (WES) analysis, Medical genetics, Genetic disposition to disease, Genetic diseases, Genetic variants, Clinical NGS informatics

Funding: Scientific and Technological Research Council of Turkey 120E522 This work was supported by the Scientific and Technological Research Council of Turkey (No. 120E522). The funders had no role in study design, data collection and analysis, decision to publish, or preparation of the manuscript.

==============================
The discovery of low-coverage (i.e. uncovered) regions containing clinically significant variants, especially when they are related to the patient’s clinical phenotype, is critical for whole-exome sequencing (WES) based clinical diagnosis. Therefore, it is essential to develop tools to identify the existence of clinically important variants in low-coverage regions. Here, we introduce a desktop application, namely DEVOUR (DEleterious Variants On Uncovered Regions), that analyzes read alignments for WES experiments, identifies genomic regions with no or low-coverage (read depth < 5) and then annotates known variants in the low-coverage regions using clinical variant annotation databases. As a proof of concept, DEVOUR was used to analyze a total of 28 samples from a publicly available Hirschsprung disease-related WES project (NCBI Bioproject: https://www.ncbi.nlm.nih.gov/bioproject/?term=PRJEB19327), revealing the potential existence of 98 disease-associated variants in low-coverage regions. DEVOUR is available from https://github.com/projectDevour/DEVOUR under the MIT license.

Introduction

Whole-exome sequencing (WES) is a technique for sequencing the protein-coding (i.e., exonic) regions of genes in a genome. WES is typically utilized to identify genetic variants that are associated with clinical phenotypes such as observable traits of a disease. As a result, WES has been widely utilized in both academic research and clinical diagnosis. Although WES is a cost-effective and convenient approach that has enabled the detection of clinically significant (e.g., deleterious) variants in almost all coding regions of the genome (Tetreault et al., 2015), it has some limitations in analysis processes (Bergant et al., 2018).

In the context of WES analysis, the presence of variants in low-coverage regions poses a challenge for accurate variant detection and interpretation. Low-coverage regions are areas of the genome with no or low sequencing coverage, resulting in a lack of read depth to confidently identify genetic variants. Read depth refers to the number of sequencing reads that align to a specific genomic region. Regions with insufficient coverage have a significantly lower number of reads compared to adequately covered regions. By analyzing the read depth across the exome, it is possible to identify regions with low-coverage.

In the literature, several studies have reported that a substantial proportion of the genome may remain low-coverage in WES experiments. For example, Bick et al. (2012) observed that approximately 20% of the exome was not adequately covered in their study cohort. Similarly, Ku et al. (2012) reported that around 15% of the exome was poorly covered in their analysis. These findings suggest that a significant portion of the exome may be susceptible to being classified as low-coverage regions. It is important to note that the number of variants expected in low-coverage regions can vary depending on the sample population, sequencing platform, exome capture kit used, and other technical factors.

The presence of low-coverage regions in whole-exome sequencing (WES) experiments can be attributed to various factors. One primary factor is the design limitations of exome capture kits (Kong et al., 2018). These kits are designed to target specific genomic regions of interest, typically focusing on protein-coding exons. However, certain genomic regions, such as those with high GC content or segmental duplications, may be challenging to capture effectively, leading to reduced coverage and potentially missed variants (Choi et al., 2009; Clark et al., 2011; Sheppard et al., 2018).

In a study conducted by Pengelly et al. (2020), the performances of two commercial exome capture kits were compared for clinical diagnostics. The researchers reported achieving 76% and 91% coverage for 100X experiments using these kits. Additionally, at 20X resolution, different exome capture kits were capable of covering 98% of the targeted genomic regions, and at 30X resolution, the coverage dropped to 97% of the genome. These findings emphasize the impact of coverage depth on the ability to capture genomic regions of interest and underscore the need to carefully consider the trade-offs between coverage depth and cost in WES experiments.

Although these exome capture kits achieved high coverage levels for most regions, the remaining uncovered regions warrant careful consideration, especially in clinical diagnostics, to ensure comprehensive variant detection and accurate genetic analysis. Low-coverage regions may contain clinically relevant variants, and their accurate identification is crucial for making informed clinical decisions.

In addition to the capture kit limitations, the lack of coverage in specific regions can also be influenced by technical factors and library preparation protocols. Sequencing biases, such as preferential amplification of certain genomic regions or biases introduced during library construction, can contribute to uneven coverage and subsequently affect variant detection (Gnirke et al., 2009; Ross et al., 2013).

Furthermore, the presence of low-coverage regions may also arise from biological factors. For example, segmental duplications and repetitive elements in the genome can hinder accurate mapping of reads, resulting in zero or low-coverage in those regions (Kiezun et al., 2012).

Therefore, the detection of low-coverage regions containing clinically significant variants, especially when they are relevant to the clinical phenotype of the patient, is of utmost importance and calls for further validation protocols such as deep sequencing. Hence, there is a need for tools to alert clinicians regarding the risks associated with low-coverage regions that are directly related to the patient’s health.

To address the challenge of identifying genomic regions with no or low-coverage in WES data, we introduce DEVOUR, a desktop application specifically designed for this purpose. Unlike traditional variant callers such as HaplotypeCaller (McKenna et al., 2010), Platypus (Rimmer et al., 2014), and Freebayes (Garrison & Marth, 2012), which focus on variant calling, DEVOUR takes a unique approach by prioritizing variant annotation and interpretation. DEVOUR is the first of its kind in this regard, aiming to identify low-coverage regions and annotate clinically important variants within them using clinical variant annotation databases.

By combining coverage analysis and comprehensive variant annotation, DEVOUR provides a powerful solution to uncover and analyze clinically significant variants, even in challenging low-coverage regions. This pioneering approach makes DEVOUR a valuable tool for researchers and clinicians in the pursuit of accurate clinical diagnosis.

In this study, we demonstrate DEVOUR’s capabilities using publicly available Hirschsprung disease-related WES data provided by Gui et al. (2017) as an example. Hirschsprung disease is a congenital disorder caused by the absence of ganglion cells in the colon, and it is known to be caused by mutations in several genes, including RET, EDNRB, and SOX10 (Tang et al., 2023).

According to the ClinVar database (Landrum et al., 2020), there are total of 32 pathogenic and likely pathogenic variants related to Hirschsprung disease. These variants are mainly clustered on chromosomes 4 and 10, with additional variants present on chromosomes 1 and 22, as depicted in Fig. 1.

Figure 1 Distribution of pathogenic and likely pathogenic variants related to Hirschsprung disease in ClinVar database.

The figure illustrates the distribution of 32 pathogenic and likely pathogenic variants associated with Hirschsprung disease in the ClinVar database. Each chromosome is represented by a different color. The majority of the pathogenic variants are clustered on chromosomes 4 and 10.

By employing DEVOUR’s coverage analysis and variant annotation approach, we successfully identified low-coverage regions in Hirschsprung disease-related WES data. DEVOUR’s comprehensive analysis allowed us to annotate clinically significant variants within these regions, providing valuable insights into potential disease-causing mutations. This capability enhances the accuracy of WES data analysis and demonstrates the significance of DEVOUR in uncovering crucial genetic information for clinical diagnosis.

Materials & Methods

DEVOUR works with a set of inputs; a read alignment file (SAM or BAM) for the WES experiment, a Browser Extensible Data (BED) file for the regions targeted by the WES capture kit, a read depth threshold to identify low-coverage regions, a human reference genome version (i.e., hg19 or hg38), and a list of annotation resources for clinically significant (e.g., deleterious) variants. DEVOUR works in three main steps as represented in Fig. 2.

Figure 2 The representation of DEVOUR’s workflow.

The first step identifies low-coverage genomic regions in a WES experiment. To this end, a user-provided read alignment file (SAM or BAM) is sorted and indexed using Samtools (Danecek et al., 2021). Per-base depth calculation is performed using Mosdepth (Pedersen & Quinlan, 2018) which is a command-line tool for calculating the sequencing coverage. During the depth calculation process, an exome capture file is used to limit the calculation of low-coverage regions to the ones targeted by the respective exome capture kit. Next, a selected read depth threshold (default = 5) is used to identify genomic regions with low-coverage. The output of this step is a list of low-coverage genomic regions in BED format. It is important to note that setting a read depth threshold of zero in DEVOUR leads to the identification of exclusive regions with absolutely no coverage. This threshold allows for the specific detection of regions that lack any sequencing reads, highlighting areas where no genetic information is captured. The human reference genome release name is specifically required in DEVOUR to accurately present users with the appropriate versions of the annotation sources. It is important to note that while the reference genome release name is necessary, users do not need to provide the actual reference genome FASTA file. DEVOUR utilizes the reference genome release name to ensure that the corresponding annotation sources aligned to the specific genome version are made available during the analysis process. This allows users to select the correct and relevant annotation sources for accurate variant annotation and interpretation. The user interface developed for the first step is shown in Fig. 3.

Figure 3 The illustration of the user interface for providing the parameters: input files (an alignment in SAM or BAM format and an exome capture file in BED format), depth threshold and human genome reference version.

The initial stage in DEVOUR’s analysis pipeline is to identify potentially uncovered or low-coverage genomic regions. A list of low-coverage genomic regions in BED format is the result of this stage.

The second step aims to enrich the low-coverage regions with variant annotations as a step toward assessing the clinical significance of variants found in these regions. Variant annotations can either come from custom in-house annotation sources or clinically significant variant databases provided by ANNOVAR (Wang, Li & Hakonarson, 2010) such as ClinVar (Landrum et al., 2020). To ensure compatibility with DEVOUR, a custom annotation database must be provided as a BED-like file. This file format should consist of four tab-delimited columns, arranged in the specified order. The columns should include the chromosome name, start coordinate, end coordinate, and variant annotation in free text. DEVOUR has configuration screens to handle the incorporation of these custom annotation sources as shown in Fig. 4. Similarly, DEVOUR has implemented mechanisms to fetch disease-specific variant databases from ANNOVAR repositories and transform them into BED-like formats like custom annotation sources. Once the annotation sources are properly configured in DEVOUR, the analysis process presents the user with a comprehensive list of available databases. This list includes all the configured annotation sources, allowing the user to easily select the desired databases for the analysis. The availability and visibility of these databases during the analysis process enhance the user’s ability to make informed decisions and effectively utilize the relevant annotation sources within DEVOUR.

Figure 4 The illustration of user interfaces developed for users to create an annotation library.

Either custom annotation sources or ANNOVAR’s disease-associated variant databases, like ClinVar, can be used to annotate variants. A BED-like file with four tab-delimited columns holding the chromosomal name, start coordinate, end coordinate, and the variant annotation in free text, in that order, can serve as the custom annotation source. To handle the inclusion of various unique annotation sources, DEVOUR offers settings panels. Similar to custom annotation sources, DEVOUR has created methods to retrieve disease-associated variant databases from ANNOVAR repositories and convert them into BED-like formats.

Variants located in the low-coverage regions from the previous step are annotated using the selected variant annotation sources. For this annotation, DEVOUR utilizes an overlap computation algorithm leveraging the interval trees; a data structure that allows for efficient computation of intervals that overlap with a query interval. In this step, the genomic coordinates for clinical annotations are stored in an interval tree data structure per chromosome and queried with the low-coverage regions from the previous step. The output of this step is a set of tables where each table contains seven tab-delimited columns; the chromosomal location for the low-coverage regions (chromosome name, start/end coordinates), depth of the region, the chromosomal location for the annotated variant (start/end coordinates), and detailed annotation retrieved from respective source. The user interface developed for the first step is shown in Fig. 5.

Figure 5 The illustration of the user interface for selecting the desired annotation source(s).

At this stage, the annotation resources in the DEVOUR library, which were prepared according to the human reference genome version selected in the previous stage, are listed.

The final step provides files to assist clinical diagnosis highlighting the variants in low-coverage genomic regions with clinical significance (e.g., pathogenicity). DEVOUR helps users to preview and export each annotation-based table from the previous step in TSV or Excel format for inspection as shown in Fig. 6.

Figure 6 The illustration of the user interface for reviewing the results.

This stage seeks to generate result files to aid clinical diagnosis by indicating mutations in genomic regions with low-coverage that have clinical significance. DEVOUR enables users to evaluate each annotation-based table from the previous stage by previewing and exporting it in TSV or Excel format.

DEVOUR is developed using the Electron framework (http://www.electronjs.org). In addition, DEVOUR has some prerequisites that must be installed; Samtools, Mosdepth, ANNOVAR, and some Python libraries; intervaltree (https://pypi.org/project/intervaltree/), pandas (Reback et al., 2022), and openpxyl (https://openpyxl.readthedocs.io/en/stable/). As part of DEVOUR installation process, the paths for an application working directory and the prerequisite applications need to be configured.

Our DEVOUR tool was benchmarked using a publicly available WES project conducted by Gui and colleagues (Gui et al., 2017), focusing on Hirschsprung disease, a congenital abnormality characterized by the absence of nerves in portions of the intestine. This WES project (NCBI Bioproject: https://www.ncbi.nlm.nih.gov/bioproject/?term=PRJEB19327) contains a total of 72 samples obtained from two sequencing platforms (Illumina and ABI SOLiD) from the Sequence Read Archive (Leinonen et al., 2011). For testing purposes, we limited our analysis to 28 paired-end samples obtained from the Illumina platform, as we did not have access to the exome capture kit information for the samples obtained from the ABI SOLiD platform. The samples analyzed on the Illumina platform had a mean coverage of 27.9X, indicating that, on average, each base in the target region was sequenced about 28 times. Moreover, approximately 74% of the bases in the target region had a sequencing coverage greater than 10 times (Gui et al., 2017).

To conduct our analysis, we acquired the FASTA files for each sample and performed sequence alignment against the reference human genome (build hg19), using HISAT2 alignment tool (Kim et al., 2019), to obtain the corresponding BAM files. Subsequently, we processed these BAM files individually using DEVOUR. The first supplementary file provides a detailed description of the analysis process for one of the samples, specifically NCBI SRA: ERR1840777 (see Supplemental File 1). This example serves to showcase the usage of DEVOUR and provides step-by-step instructions for our analysis.

To identify clinically significant annotations in low-coverage regions (read depth < 5) and high-coverage regions (read depth >= 5) of these samples, we fed the BAM files to DEVOUR along with the exome capture kit used by the Illumina sequencing platform (Illumina, San Diego, CA, USA).

Results

DEVOUR was used to identify ClinVar annotations overlapping with the identified low and high-coverage regions. In our analysis of 27 samples using DEVOUR, we detected at least one Hirschsprung-related pathogenic variant in high-coverage regions, as shown in Fig. 7 (see Supplementary File 2). On average, we identified approximately 17 pathogenic variants per sample, with a standard deviation of ±6.

Figure 7 The distribution of samples containing Hirschsprung-related pathogenic variants on low- and high-coverage regions.

DEVOUR analysis (with the default read depth threshold) revealed at least one Hirschsprung-related pathogenic variant in low-coverage regions (read depth < 5) and high-coverage regions (read depth ≥ 5) for 18 and 27 out of 28 samples, respectively. Sample ERR1840777 is distinctive as the only sample containing Hirschsprung-related pathogenic variants exclusively in low-coverage regions.

Furthermore, in 18 of the samples, we found at least one Hirschsprung-related pathogenic variant located in low-coverage regions (see Supplementary File 3). The average number of such pathogenic variants identified per sample was approximately 8, with a standard deviation of ±6. In one sample (NCBI SRA: ERR1840777), no Hirschsprung-related pathogenic variants were identified in high-coverage regions. However, using DEVOUR, we detected a total of 27 Hirschsprung-related pathogenic variants in low-coverage regions within this sample. Notably, 25 out of these variants were located in regions with no coverage, indicated by a depth value of 0 (see Supplementary File 4). To validate our findings, we utilized the NCBI Sequence Viewer (Rangwala et al., 2021) to map the sequence reads obtained from sample ERR1840777 to the reference genome, allowing us to assess the coverage. Figure 8 illustrates the sequence coverage on chromosome 10 between coordinates 43,000,000 and 44,000,000 for this sample. The regions without bars indicate low or no read coverage, which further supports the presence of low-coverage regions in the dataset. This visual representation enhances the comprehensibility of our identified low-coverage regions and bolsters our findings.

Figure 8 Sequence coverage on chromosome 10 (Coordinates: 43,000,000 - 44,000,000) for sample ERR1840777.

The figure displays the sequence coverage on chromosome 10 in the genomic region spanning coordinates 43,000,000 to 44,000,000 for sample ERR1840777. Regions without bars on the graph indicate low or no read coverage, highlighting the presence of low-coverage regions in the dataset. The visual representation provides valuable insights into the distribution of sequence coverage and confirms the identification of low-coverage regions in the sample, supporting the findings of this study. NCBI Sequence Viewer Link: https://www.ncbi.nlm.nih.gov/projects/sviewer/?id=CM000672.1&tracks=[key:sequence_track,name:T378820,display_name:Sequence,id:T378820,dbname:GenBank,annots:NA,ShowLabel:false,ColorGaps:false,shown:true,order:1][key:alignment_track,name:ERR1840777,display_name:ERR1840777,id:STD2123359385,dbname:SRA,setting_group:cSRA,annots:ERR1840777,Layout:Adaptive,StatDisplay:15,Color:ShowDifferences,UnalignedTailsMode:glyph,HideSraAlignments:none,sort_by:,LinkMatePairAligns:false,ShowAlnStat:true,AlignedSeqFeats:false,Label:false,IdenticalBases:false,shown:true,order:7]&srz=ERR1840777&assm_context=GCA_000001405.3&mk=42833500|42833500|blue|9&v=43251931:43680986&c=FFFFFF&select=null&slim=0..

To gain further insights from DEVOUR’s analysis, we conducted an evaluation of the total length of low-coverage regions per chromosome for each sample (see Supplementary File 5). Our analysis revealed that samples with Hirschsprung-related variants tend to exhibit larger proportions of low-coverage regions. This observation suggests the need for repeating WES experiments for these specific patients/samples to enhance coverage and improve the accuracy of variant detection.

An illustrative example is sample ERR1840777, wherein Hirschsprung-related variants were identified exclusively in low-coverage regions. In this scenario, DEVOUR’s analysis serves as a valuable guide for clinicians, directing them to focus on these specific chromosome regions through more in-depth approaches such as deep sequencing. By conducting targeted experiments, clinicians can ascertain the presence of genuine variants and achieve precise clinical interpretations. Unfortunately, NCBI SRA did not provide any clinical phenotype for this sample, but regardless, our finding asks for further investigation to improve or potentially correct this sample’s Hirschsprung disease genotype.

The computation for each sample took eight minutes on average (min: 4 min, max: 12 min) on an Intel i7-5820K based virtual machine with 10 GB memory. These results show the importance of inspecting low-coverage regions using alternative methodologies such as deep sequencing as these regions may contain variants potentially critical for the clinical diagnosis. Without DEVOUR, the potential existence of such variants would not have been possible to identify and may result in missing potential diagnoses.

With DEVOUR’s ability to incorporate these extended databases seamlessly, it becomes even more valuable in facilitating precise and comprehensive variant analysis, aiding in the identification of pathogenic variants, and contributing to improved diagnostic outcomes for patients. As the number of clinically important variants available in both public and private variant annotation resources continues to increase over time, we anticipate that DEVOUR will become increasingly beneficial, particularly for undiagnosed patients. The expanding databases of clinically significant variants provide a valuable resource for accurate variant interpretation and diagnosis.

Conclusions

In the context of whole-exome sequencing (WES) analysis, it is crucial to identify and notify clinicians about clinically significant variants located in low-coverage regions to avoid missing potential diagnoses. Low-coverage regions can contain important genetic variants that contribute to the observed clinical phenotype but may go undetected due to insufficient sequencing coverage. By identifying these variants, clinicians can gain valuable insights into the underlying genetic basis of the patient’s condition and make more informed diagnostic and treatment decisions. Therefore, it is essential to develop tools and strategies that effectively address the challenge of detecting and interpreting variants in low-coverage regions to maximize the diagnostic yield of WES analysis.

To address this need, we have developed DEVOUR, a desktop application that facilitates the analysis of WES experiments and identifies clinically significant variants in regions with low or no coverage. DEVOUR serves as a valuable addition to WES analysis pipelines, particularly those focused on detecting and annotating variants in covered genomic regions.

Looking towards the future, we envision DEVOUR’s expansion to encompass not only ClinVar but also central protein databases such as Uniprot, as well as established mutation databases like HGMD. This evolution is expected to significantly augment DEVOUR’s versatility, extending its applicability to both germline studies and somatic analyses. The anticipated outcome is enhanced robustness, particularly in scenarios where pinpointing precise target points is of utmost importance, such as in somatic investigations. Moreover, our vision includes the potential extension of DEVOUR’s capabilities to encompass whole-genome sequencing (WGS) experiments, thereby broadening its scope to cater to an even broader array of genomic analyses. The pipeline integrated into DEVOUR is designed to accommodate WGS data seamlessly. Users are advised to provide a BED file that encompasses the entire genomic coordinate instead of solely exonic regions when working with WGS data. This BED file, generated manually by including the desired genomic regions for analysis, ensures effective utilization of DEVOUR’s comprehensive variant annotation and interpretation capabilities in WGS data analysis.

By leveraging DEVOUR, researchers and clinicians can enhance their understanding of genomic variants, enabling more accurate and informed decision-making in clinical settings. With its versatility in handling both WES and potentially WGS data, DEVOUR is a valuable tool for comprehensive variant analysis and interpretation in a clinical research setting.

Supplemental Information

Supplemental Information 1 Step-by-step visualization for an example DEVOUR run

An example run for the SRA sample ERR1840777.

Click here for additional data file.

Supplemental Information 2 The list of Hirschsprung-related pathogenic variants located on high coverage regions of 27 samples

Each row in this file provides the SRA sample id, chromosome name, start/end chromosomal coordinates for the located region, coverage depth for the region, start/end coordinates for the ClinVar annotation and the annotation description.

Click here for additional data file.

Supplemental Information 3 The list of Hirschsprung-related pathogenic variants located on low coverage regions of 18 samples

The SRA sample id, chromosome name, start/end chromosomal coordinates for the located region, coverage depth for the region, start/end coordinates for the ClinVar annotation and the annotation description.

Click here for additional data file.

Supplemental Information 4 The list of Hirschsprung-related pathogenic variants located on the ERR1840777 sample

Additional information, ClinVar variation id, and gene name(s), for the located variants.

Click here for additional data file.

Supplemental Information 5 Evaluation of total length of low-coverage regions per chromosome for each sample

The data highlights the distribution of low-coverage regions across different chromosomes and their respective lengths for each sample. Samples with Hirschsprung-related variants are observed to exhibit larger proportions of low-coverage regions, indicating potential challenges in accurately detecting variants in these genomic regions. Please note that if a sample has a length of 0 for a particular chromosome, it means there were no records for that chromosome in the sample. This could be due to various factors, such as technical limitations or the absence of reads in that specific chromosome for the given sample.

Click here for additional data file.

We would like to acknowledge that this study is part of Erdem Türk’s Ph.D. thesis at Muğla Sıtkı Koçman University Graduate School of Natural and Applied Sciences.

Additional Information and Declarations

Competing Interests

Author Contributions

Data Availability

The authors declare there are no competing interests.

Erdem Türk conceived and designed the experiments, performed the experiments, analyzed the data, prepared figures and/or tables, authored or reviewed drafts of the article, and approved the final draft.

Akif Ayaz analyzed the data, authored or reviewed drafts of the article, and approved the final draft.

Ayhan Yüksek performed the experiments, prepared figures and/or tables, contributed to the development of the tool, and approved the final draft.

Barış E. Süzek conceived and designed the experiments, analyzed the data, authored or reviewed drafts of the article, and approved the final draft.

The following information was supplied regarding data availability:

DEVOUR is available at GitHub: https://github.com/projectDevour/DEVOUR.

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
