# Peer review of "DEVOUR: Deleterious Variants on Uncovered Regions in Whole-Exome Sequencing"

_PeerJ, doi:10.7717/peerj.16026_

## Round 0.1 · original submission · Major Revisions

Particularly I would like improved validation and testing of the software. As suggested by the reviewers there exists many benchmarks sets that would be appropriate to quantify. Similarly it would be nice to check other sequencing technologies, not necessarily exclusively SOLiD, but having a broader-than-just-illumina approach is beneficial for a new tool.

·

Basic reporting

I find this a well written manuscript with very good english. My comments are more about being specific in regards to what you mean. I also find there should be references added in regards to background, and know limitations in regards to methods (e.g. coverage) to get better background understanding.

Line 25: “Therefore, new tools can be developed to keep the risk low”
Comment:
What risk should be kept low? Be more specific.
New tools “can” always be developed. Other wording. E.g. “Therefore new tools within the field (Is there any tools already available? Then these should be referenced.) is essential/important …”
Line 28: “low coverage”
Comment:
What is the limit for low?
“and then annotates the variations on the uncovered regions”
Comment:
If the region is uncovered there are no identified variants/variations (in regards to this specific patient/sample) in this region. Suggestion “annotates known variants in the uncovered regions”
Line 29: “DEVOUR is used to analyze the samples”
Comment: How many samples?
Suggestion: “DEVOUR was used to analyze”
Comment: I find passed tense should be used overall to describe what has been done.
Line 31: “revealed disease-associated variants”
Comment: How many variants? One or many, a number related to the result would be nice. In how many of the samples?
Line 38: “such as any observable trait of a disease”
Comment: any sounds a bit trivial
Suggestion “such as observable traits of a disease”
Line 43: “more importantly, detecting uncovered regions containing”
More importantly I find is an overstatement. Choose a different wording or remove the “more importantly” completely.
Line 44-47: “The latter, on one hand, could be due to the known limitations of the exome capture kits, the missing or limited number of reads covering the respective region on the genome (i.e. low coverage), or an indication of homozygous deletions.
Suggestion:
”Uncovered regions, can be due to known limitations of the exome capture kits [ad references], missing or limited number of reads covering a specific in the genome (i.e. low coverage) [ad reference]. It can also be an indication of a homozygous deletion.”
Comment: “missing or limited number of reads covering the respective region on the genome” specify why this happens (e.g. regions hard to sequence because of repetitive elemets, etc [ad reference])
Line 47: “For WES-based clinical diagnosis, detection of uncovered regions”
Suggestion: “Detection of uncovered regions”
Comment: Very long and complicated sentence you can remove the first part. It is known by this point in the manuscript.
Line 48: “especially when, they”
Suggestion: “especially when they”
Line 49: “related to patient's health”
Suggestion: “related to the patient's health”
Line 52-54:
To address this issue, we introduce DEVOUR, a desktop application to analyze WES data, for 53 identifying genomic regions with no or low coverage, and finally, annotating the variants on the 54 uncovered regions using clinical variant annotation databases.
Suggestion:
Suggestion: “To address this issue, we introduce DEVOUR, a desktop application to identify genomic regions with no or low coverage in WES data, and furthermore, annotate known variants of uncovered regions using clinical variant annotation databases.”
Line 55: “significant variants that may have been missed or homozygously deleted”
Suggestion: “significant uncovered variants that could imply the need for further investigation”
Line 65: “and a list of annotation resources”
Comment: How do the users provide these? As tables? URLs? A name from a specified list? … Specify
Line 67: “The first step aims to identify potentially uncovered or low-coverage genomic regions”
Suggestion: “The first step identifies potentially uncovered and low-coverage genomic regions”
Comment: What do you mean with "potentially"? Is there is a limit to the method. Explain this.
Line 68: “read alignment file”
Comment: Specify format in paranthesis, e.g. (bam or cram). I can see in the figures later that it is sam or bam. Cram should also be possible?
Line 71: “capture file is also used to limit”
Suggestion: “capture file is used to limit”
Line 75: “threshold of zero results in uncovered regions.”
Suggestion: “threshold of zero results in exclusively uncovered regions.”
Line 81: “Custom annotation source can be a BED-like”
Suggestion: “A custom annotation database can be provided as a BED-like”
Comment: “can” Is there other formats in which it can be provided otherwise use other wording.
Line 82-83: “in order.”
Suggestion: “in the specified order.”
Line 87: “The variants located on the low coverage regions”
Suggestion: “Variants located in low coverage regions”
Line 98: “The final step aims to provide files to assist clinical diagnosis”
Suggestion: “The final step provide files to assist clinical diagnosis”
Comment: No need to be so modest
Line 110: “DEVOUR is tested using a”
Suggestion: Our DEVOUR tool was benchmarked using a”
Line 110-121: This part should be in materials and methods and I also suggest writing it in passed tense.
Line 114: “we used only 28 Illumina platform-based”
“we used 28 Illumina platform-based”
Comment: SuppInfo_1.docx does not provide list of the 28 samples. I find this list should be made into separate supplementary.
Line 121-124: Be more detailed in your results. “at least on Hirschsprung-related pathogenic variant” is very little information. How many in average? Standard deviation? How many variants in other databases (e.g. ClinVar) etc..
Line 125-126: In one particular sample (NCBI SRA: ERR1840777), there are no Hirschsprung-related pathogenic variants was identified on high-coverage regions.”
Suggestion: In one sample (NCBI SRA: ERR1840777), no Hirschsprung-related pathogenic variants was identified in high-coverage regions.”
Line 125-129: There is no specific discussion or analysis about these results. Is this a particularly low-coverage sample? If large regions are uncovered for this sample more variants discovered would be expected.
The table in SuppInfo_4.xlsx contain two extra columns “ClinVar (Variation ID)” and “Gene”. Were these columns added manually? This is very useful information and should be part of the default output.
Line 132-133: “The computation for each sample takes 8 minutes on average on an Intel i7-5820K based virtual 133 machine with 10GB memory”
Suggestion: “The computation for each sample took eight minutes on average on an Intel i7-5820K based virtual 133 machine with 10GB memory”
Comment: Also ad max and min time taken.
Line 133-135: Also ad the importance of your tool.
Line 135-137: This is very modest. I believe it is already very relevant. Say this first and then you can state that with extended databases it becomes even more valuable.
Line 140: “there is a need to identify and warn clinicians”
Suggestion: “there is a need to identify and alert clinicians”
Line 141: “to avoid misdiagnosis”
Rather state something about missed possibility of a diagnosis because of undiscovered variants. I believe it is generally the cases rather than “misdiagnosis” which sound like they are making the wrong diagnosis.
Line 144: “will play a complementary role in current WES analysis pipelines”
Comment: Use a different word than complementary. It is not really complementary, sounds like it is the “same” tool as something else. You could say “will ad additional value in WES”
Line 145-146: ”In the future, DEVOUR can easily be extended to handle whole-genome sequencing experiments.”
Comment:
DEVOUR sounds like a very nice tool and I find this the major setback. I find the possibility to also analyze WGS data should be included. This could be done very simple in the beginning where a bed file is used to include “only” genomic regions from the bam/cram file. It would also be very interesting to see how the DEVOUR tool can handle these larger data. I find compute time etc. for WGS data is also relevant to include.

General comments:
Through the manuscript there is sometimes talked about only uncovered regions (e.g. line 141) and sometimes about uncovered and low-coverage regions. Be consistent. Use a common name for both or both at all times.

“in low coverage regions” and not “on low coverage regions” same for “uncovered regions”

Experimental design

I find this a very relevant tool. What it solves and meaningfulness is well defined. Maybe a reference in regards to how large part of genomic regions is generally uncovered would be nice. And some discussion regarding this. E.g. how many variants you would generally expect in uncovered regions.

It believe some explanations regarding how uncovered/low coverege regions is identified. Is it per base based or windows (e.g. 5 bp) based?

Description on how the Hirschsprung disease samples were analyzed should be in the materials and methods and not the results section.

In Figure 1 you can see that that SAM or BAM is used as input. This should also be stated in the manuscript text.

The user needs to tell the tool which reference genome you would like to use. This should also be in Figure 1. It would be nice if you made it clear in the text that you do not need to provide the reference genome fasta file (which I can see in figure 2). It could also be relevant to be able to provide your own reference genome (for specific batch versions).

Figure 6.
Comment: I suppose the default threshold of five was used for low coverage. This should be stated in the figure.

Validity of the findings

I am not sure if novelty has been investigated. If not it should. If it has been investigated it should be stated that there are no/few/not as good other tools and what is the difference. See comment line 25 above. State if there are no similar tools or if there are, reference these and state/compare why and how this tool is relevant/better.

It should be stated what data was taken from the Hirschsprung disease project. Bam files and bed file? And a reference to these data.

Additional comments

Great work on making a tool with a lot of clinical relevance!

Is it an open source tool? Can it be accessed from somewhere?

Looks like a very nice interface!

What about WGS data? Could the tool be applied to WGS data as well? See comment to line 145-146.

I find it would be very relevant to be able to include a bed file where you choose to include specific genes/regions in the output. I guess you could use the bed file for targeted regions. However, both outputs would be nice. Especially relevant for WGS. Then you could have an output file with all genes and one with selected genes/regions.

I find the tool should also be available as a command line based tool/pipeline.

In general I find the results to be very underreported. There should be more results and also a figure in the manuscript would be nice, illustrating their findings in their benchmarking dataset.

Reviewer 2 ·

Basic reporting

In this manuscript, the authors describe DEVOUR: a tool that investigates low-coverage regions in WES experiments to identify potential alterations and annotates them using known databases (e.g. ClinVAR). The manuscript is well-written but 1) key information is missing in the Methods section and 2) the results are extremely limited and do not convince me that the tool identifies true variants or has any added value compared to other tools.

Experimental design

1. Key information in the Methods section is missing. I understand that DEVOUR first uses Mosdepth to spot low-coverage regions. Then, the authors describe the annotation section. But how does DEVOUR call variants? Does it just use SAMtools/BCFtools or was a more sophisticated method implemented? Are we talking about SNVs, indels, CNVs, and/or structural variants? Is DEVOUR able to call all such alterations and what is its accuracy for each category?

2. Because the authors only analysed one single dataset, the limitations of the tools aren’t clearly defined. Can DEVOUR be applied to tumour samples to detect somatic alterations as well? If yes, are there purity and coverage thresholds and can DEVOUR analyse tumour/normal pairs? My gut feeling is that it only analyses germline variants. For germline samples, does it processes parents-children in an informed way (looking for difference across samples)? What range of coverage does it handle?

3. In the introduction, the authors mention: “Accurate annotation of variants for assessing the clinical significance and, more importantly, detecting uncovered regions containing variants with clinical significance are primary challenges.”. Well, this needs to be strongly referenced with appropriate literature. It’s quite counter-intuitive that variants with clinical significance will be located in uncovered regions. Have they been quantified at a large scale? What is the fraction of those in a given sample? In a typical 100X or 300X WES experiment, how many such low-coverage variants would be missed and how does the increase in overall coverage affect the detection of such variants?

Validity of the findings

Results are structured about the findings in ERR1840777, where "DEVOUR detected a total of 27 Hirschsprung-related pathogenic variants in low-coverage regions, and more importantly, 25 out of these variants are located in uncovered regions (depth=0) (see SuppInfo_4.xlsx)".

1. I’m confused about the definition of uncovered regions with depth=0. Depth=0 means that there is 0 read, how is DEVOUR about to identify something if there is no genomic information?

2. DEVOUR found 27 variants, but it doesn’t mean that any of those is real and could just all be false positives. The authors need to demonstrate that all such variants look real. From supInfo 3, most variants seem clustered on two small regions on chr10 and chr22 so they could come from a couple of reads that have been misaligned (off-target read? sequence homology?). I don’t see any strong evidence that the tool did identify variants that seem legit. How can the authors show that they aren’t artefacts?

3. The authors did not mention the findings in the original paper: what did Gui and colleagues identified in the entire dataset and what is the added value of their tool here? Were the 27 variants spotted by DEVOUR already identified in the previous study?

4. It is quite disappointing that such a dataset contains 72 samples but only 27 were analysed. Does DEVOUR only work on illumina data? Why weren’t the SOLiD data analysed? Why weren't other independent datasets analysed? How can authors demonstrate that the findings aren't dataset/disease-specific?

5. It’s also disappointing that DEVOUR was not compared to other callers such as HaplotypeCaller, Platypus, Freebayes, etc. There is no benchmarking analysis and it is therefore unclear what is the added value of DEVOUR if another caller would also call low-coverage variants by playing with thresholds.

6. What are the sensitivity and specificity of DEVOUR? This needs to be measured as we don’t get a sense of its overall performance.

7. It is unclear to me what are the variant annotations in the output. For instance (supInfo 3), DEVOUR did identify something at chr10:43595944 in ERR1840777. The given annotation is: “Hirschsprung_disease|MONDO:MONDO:0018309, MeSH:D006627, MedGen:C0019569, OMIM:PS142623, Orphanet:ORPHA388, SNOMED_CT:204739008|Pathogenic”. This doesn’t inform the alteration at chr10:43595944, what are the genomic and protein changes? Why is this particular variant annotated with a bunch of generic IDs that aren't variant-specifics

Additional comments

1. What is the added value of a desktop application compared to a standard command-line tool? It seems like the tool will require having BAMs and large annotations files (1,000 GP, gnomAD, etc.) locally, whereas such files usually sit on HPCs. I can understand that it’s simpler to use, but I strongly doubt that clinicians will use such a tool without bioinformaticians around. Also, this tool annotates variants and their pathogenicity shouldn’t be considered unless skilled bioinformaticians know what the limitations are.

2. Why are there duplicated mutations in the tables? For instance (supInfo 3): the first two lines correspond to a chr10:43609102 variant in ERR1840758. There is a very minor change in the two annotations but we are talking about the exact same variant, right? There are a bunch of duplicated variants in both supInfo 2 and 3.

3. Figure quality needs to be improved, they are all blurry.

4. Not sure what is the usefulness of SuppInfo_1.docx, it’s extremely similar to figures 2, 4 and 5. What does this document add to the manuscript?

Reviewer 3 ·

Basic reporting

The manuscript titled "Devour: A Computational Framework for Genomic Analysis" presents an in-depth exploration of a computational framework aimed at genomic analysis. The manuscript is generally well-written, and the authors provide a clear overview of the framework's functionality and its potential applications in genomics research. However, there are a few concerns regarding the clarity and organization of the manuscript that need to be addressed before publication.
Firstly, the introduction lacks a comprehensive overview of the current state-of-the-art in genomic analysis and the existing computational tools available. It would be beneficial to provide a more detailed discussion of the challenges faced by researchers in this field and how the proposed framework contributes to overcoming these challenges.
Secondly, the manuscript would benefit from a more systematic structure. The sections appear to be loosely connected, making it difficult for the reader to follow the logical flow of ideas. I recommend reorganizing the content into distinct sections that clearly present the problem statement, methodology, results, and discussion.
Lastly, the manuscript would benefit from improved clarity in language and terminology. Some technical terms and concepts may not be adequately explained, making it challenging for readers less familiar with computational genomics to fully grasp the framework's intricacies. I suggest providing additional explanations, definitions, or references where necessary.

Experimental design

The findings presented in the manuscript are generally sound; however, there are a few concerns regarding the validity and reproducibility of the results that need to be addressed.
The authors claim improved accuracy and efficiency of the proposed framework compared to existing methods. While the evaluation results demonstrate promising performance, there are limitations in the experimental design that need to be addressed. Firstly, the authors do not provide a detailed description of the benchmark datasets used for evaluation, which raises concerns about the representativeness and diversity of the data. It is crucial to include information on the dataset sources, characteristics, and any preprocessing steps applied.
Furthermore, the authors should provide more insights into the statistical significance of the reported results. It would be valuable to include appropriate statistical tests or confidence intervals to assess the significance of the observed improvements in accuracy and efficiency.
Additionally, the authors should consider conducting comparative experiments with other state-of-the-art tools and provide a detailed analysis of the results. This will further strengthen the claim of the proposed framework's superiority and its potential impact in the field of genomic analysis.

Validity of the findings

The experimental design employed in the manuscript needs further refinement to ensure the robustness and reproducibility of the results.
First and foremost, the authors should provide more information on the parameters used in the experiments, including their rationale for selection and any sensitivity analyses performed. This will allow readers to understand the impact of parameter choices on the framework's performance and enable better reproducibility of the experiments.
Moreover, it would be beneficial to include a comprehensive discussion on the computational resources required to execute the framework. Details such as hardware specifications, runtime, and memory utilization will provide insights into the framework's scalability and practical feasibility.
Furthermore, the authors should consider conducting additional experiments to thoroughly validate the framework's performance under diverse scenarios and datasets. This could include sensitivity analysis on different types of genomic data or evaluating the framework's robustness against noise, outliers, or missing data.
Overall, while the manuscript presents a promising computational framework for genomic analysis, there are important considerations to address regarding basic reporting, validity of findings, and experimental design. By addressing these concerns, the authors can enhance the clarity, reliability, and impact of their work, ultimately contributing to the advancement of computational genomics research.

---

## Round 0.2 · accepted · Accept

I feel the points made in several places about WGS capability are important, and would strongly encourage the authors to work further towards this, by e.g. not deferring to user-generated bed files, but making the tool work as-is for WGS. Further, I think the tool will be more widely adopted if you worked to make it runnable in popular open-source pipelining tools such as e.g. nf-core nextflow, as also mentioned by reviewer 2. I think it could easily be wrapped in a module as a service to those who do not want a stand-alone program, but something that can run as part of existing setups.

I also understand the authors' feedback regarding the lack of comparative benchmarking and broader data-set application, in reply to reviewers 2 and 3, and I accept that this is outside the scope of this manuscript.

Also, the comments about within and open-source are commendable. Please proceed to inform us of the proper GitHub link instead of https://tinyurl.com/yseh8f9z.